# Nanoparticle Delivery of Novel PDE4B Inhibitor for the Treatment of Alcoholic Liver Disease

**DOI:** 10.3390/pharmaceutics14091894

**Published:** 2022-09-07

**Authors:** Jingyi Ma, Virender Kumar, Ram I. Mahato

**Affiliations:** Department of Pharmaceutical Sciences, University of Nebraska Medical Center, 986025 Nebraska Medical Center, Omaha, NE 68198-6025, USA

**Keywords:** phosphodiesterase 4 inhibitor, alcoholic liver disease, nanoparticles

## Abstract

The incidence of alcoholic liver disease (ALD) is increasing worldwide while no effective treatment has been approved. The progression of ALD has proven to be related to the upregulation of phosphodiesterase 4 (PDE4) expression, and PDE4 inhibitors showed potential to improve ALD. However, the application of PDE4 inhibitors is limited by the gastrointestinal side effects due to PDE4D inhibition. Therefore, we used a novel PDE4B inhibitor KVA-D88 as the therapeutic for ALD treatment. KVA-D88 inhibited inflammatory response, promoted β-oxidation, increased the level of antioxidants in the hepatocytes, and suppressed hepatic stellate cell (HSC) activation in vitro. To improve the solubility and availability in vivo, KVA-D88 was encapsulated into mPEG-b-P(CB-co-LA) nanoparticles (NPs) by solvent evaporation, with a mean particle size of 135 nm and drug loading of 4.2%. We fed the male C57BL/6 mice with a Lieber–DeCarli liquid diet containing 5% (*v*/*v*) ethanol for 6 weeks to induce ALD. Systemic administration of KVA-D88 free drug and KVA-D88-loaded NPs at 5 mg/kg significantly improved the ALD in mice. KVA-D88 significantly ameliorated alcohol-induced hepatic injury and inflammation. KVA-D88 also markedly reduced steatosis by promoting fatty acid β-oxidation. Liver fibrosis and reactive oxygen species (ROS)-caused cellular damage was observed to be alleviated by KVA-D88. KVA-D88-loaded NPs proved better efficacy than free drug in the animal study. In conclusion, the novel PDE4B inhibitor KVA-D88-loaded NPs have the potential to treat ALD in mice

## 1. Introduction

Alcoholic liver disease (ALD) is a worldwide health issue and an increasing health burden that affects around 4.7% of United States adults and is a leading indication for liver transplantation [1]. The ALD-related death rate has increased dramatically in the past ten years to 7.3 per hundred thousand [1]. The fact that currently abstinence and nutrition support are the only feasible treatment for ALD necessitates the development of new therapeutic strategies [2].

Cyclic adenosine monophosphate (cAMP) is a critical second messenger that mediates numerous intracellular biological pathways [3,4]. The phosphodiesterase (PDE) family regulates the cAMP level, degrading cAMP to AMP. PDE isozymes distribute extensively in different tissues and organs. Among them, PDE4 enzymes are specific to cAMP degradation and are highly expressed in the liver [5]. Previous studies have revealed the link between PDE4 and ALD, and inhibition of PDE4 is beneficial to ALD treatment in animal studies [6]. The application of PDE4 inhibitors is severely limited due to the low therapeutic index and gastrointestinal adverse effects, including nausea and emesis [7]. The PDE4 family is comprised of PDE4A, 4B, 4C, and 4D. Research has shown that the side effects of pan PDE4 inhibitors are the consequence of PDE4D inhibition in the central nervous system [8]. To overcome this limitation, in this study we utilized a novel small molecule compound KVA-D88 to treat ALD [9]. KVA-D88 is a potent and selective PDE4B inhibitor as a 1H-pyrrolo[2,3-b]pyridine-2-carboxamide derivative. In an in vitro test, we found that the IC50 values of KVA-D88 are 140 and 880 nM for PDE4B and PDE4D, respectively [10]. The selectivity and inhibitory efficacy of KVA-D88 was accomplished by its unique structure, which allowed *π-π* stacking with PDE4B residues and prevented cAMP from approaching the catalytic domain [9]. 

Polymeric nanoparticles (NPs) have attracted significant attention as drug carriers because of their fascinating abilities to improve delivery efficiency [11]. Encapsulating hydrophobic compounds into NPs can substantially enhance drug bioavailability, increase circulation time, decrease drug toxicity, and prevent the off-target effect [12]. The slow degradation rate of NPs can improve drug pharmacokinetics by enabling a continuous drug release [13]. To overcome the poor solubility of KVA-D88 and augment the drug availability, we employed methoxy poly(ethyleneglycol)-b-poly(carbonate-co-lactide) [mPEG-b-P(CB-co-LA)] formed NPs as the drug carrier for the treatment of ALD in vivo [14]. The therapeutic efficacy of the formulation to alleviate ALD was examined and proven to be promising by both in vitro and in vivo experiments. Our study provided insight into a novel therapeutic strategy for ALD and demonstrated its potency.

## 2. Materials and Methods

### 2.1. Materials

KVA-D88 was synthesized and provided by Dr. Corey Hopkins of the University of Nebraska Medical Center. 2,2-Bis(hydroxymethyl) propionic acid, 1,8 diazabicycloundec-7-ene (DBU), benzyl bromide, L-lactide, protease inhibitors, and phosphatase inhibitors were purchased from Sigma-Aldrich (St Louis, MO, USA). Methoxy poly(ethylene glycol) (mPEG, 5000 Da) and Laemmli buffer were purchased from Alfa Aesar (Tewksbury, MA, USA). Methylene chloride, diethyl ether, ethyl acetate, pyridine, toluene, antibiotic-antimycotic, radioimmunoprecipitation assay (RIPA) buffer, and Pierce BCA protein assay kit were purchased from ThermoFisher Scientific, Inc. (Waltham, MA, USA). ^1^H NMR spectrum was obtained using a Bruker Avance-III HD 400MHz NMR spectrometer. NMR data was analyzed using TopSpin 3.5 (Bruker, Billerica, MA, USA). Dulbecco’s Modified Eagle’s Medium (DMEM) high glucose was purchased from American Type Culture Collection (Manassas, VA, USA). DMEM/ F12 was purchased from Hyclone (Logan, UT, USA). Phosphate-buffered saline (PBS) buffer and 0.25% trypsin was purchased from Corning (Tewksbury, MA, USA). Fetal bovine serum (FBS) was purchased from Atlanta Biologicals (Flowery Branch, GA, USA) 10× Tris-Glycine SDS running buffer was purchased from Invitrogen (Waltham, MA, USA). Mini PROTEAN TGX gels were purchased from Bio-Rad Laboratories (Hercules, CA, USA). RNeasy Mini kit was purchased from Qiagen (Hilden, Germany). LightCycler 480 multiwell plate 96 and SYBR green were purchased from Roche Diagnostics (Indianapolis, IN, USA).

### 2.2. Cell Culture

Murine hepatocytes AML12 cells were purchased from the American Type Culture Collection (ATCC) (Manassas, VA, USA) and cultured in DMEM/F12 medium supplemented with 10% FBS, 1% penicillin/streptomycin, 10 µg/mL insulin, 5.5 µg/mL transferrin, 5 ng/mL selenium, and 40 ng/mL dexamethasone. RAW 264.7 cells were kindly provided by Dr. John S. Davis at the UNMC and cultured in DMEM medium supplemented with 10% FBS and 1% penicillin/streptomycin. HEK-293 cells and LX-2 human hepatic stellate cell line were cultured in high glucose DMEM medium supplemented with 10% FBS and 1% penicillin/streptomycin. All cells were cultured at 37 °C and 5% CO_2_ in an incubator.

### 2.3. Cytotoxicity Assay

The potential cytotoxicity of KVA-D88 was assessed by MTT (3-(4,5-dimethylthiazol-2-yl)-2,5-diphenyltetrazolium bromide) assay. Briefly, AML12 and LX-2 cells were plated in a 96 well plate and allowed to adhere overnight. A series of concentrations of KVA-D88 in DMSO solution was added to the cells and incubated for 48 h. At the end of the incubation, MTT solution was added to each well at a final concentration of 0.5 mg/mL and incubated for 3 h at 37 °C in the dark. The medium was removed and 200 μL DMSO was added to each well to dissolve the purple crystal. The cell viability was calculated by the absorbance at 570 nm wavelength. 

### 2.4. Measurement of KVA-D88 Inhibitory Activity In Vitro

The intracellular inhibitory activity of KVA-D88 on PDE4B and PDE4D was measured by a PDE4B and PDE4D cell-based activity assay kit and Dual Luciferase (Firefly-Renilla) Assay System (BPS Bioscience) according to the manufacturer’s instructions. Briefly, HEK-293 cells were plated in a 96 well plates and transfected with PDE4B or PDE4D expression vectors and cAMP response element (CRE) luciferase reporters before incubation with different concentrations of KVA-D88. CRE promoter driven luciferase expression was stimulated by 10 μM Forskolin for 5 h, and a dual luminescence assay was performed afterward. Luciferase intensity was measured by Molecular Devices SpectraMax M5 at 340 nm and 490 nm.

### 2.5. Measurement of cAMP Level

To measure the intracellular cAMP level, AML12 cells were plated in 100 mm dishes and allowed overnight to settle down. EtOH (50 mM) was added to cells with or without 20 min preincubation of 10 μM KVA-D88. EtOH treatment was carried out for 48 h. At the end of the treatment, cells were collected, and the intracellular level of cAMP was measured using the cAMP ELISA kit (Enzo Life Sciences, Farmingdale, NY, USA, Cat# ADI-900-163) according to the instructions.

### 2.6. Quantitative Real Time RT-PCR and Western Blot

For quantitative Real time PCR (qRT-PCR), the total RNA was isolated using RNeasy Mini kit (Qiagen) and reverse transcribed to cDNA using Taqman^®^ reverse transcription kit. qRT-PCR was performed with Roche LightCycler 480 and SYBR^®^ Green I. Primers were synthesized by IDT Inc. (Coralville, IA, USA) and are shown in Appendix A. The comparative ΔΔCt method was used for gene expression analysis.

For Western blot assay, proteins from cells were isolated with RIPA buffer supplemented with 1% protease inhibitor and phosphatase inhibitor. Isolated proteins were quantified by BCA assay and denatured with Laemmli loading buffer at 90 °C for 5 min. Denatured samples were loaded to 4 to 15% SDS–polyacrylamide gel electrophoresis gel and transferred by electroporation to polyvinylidene difluoride (PVDF) membrane. The PVDF membrane was blocked with 5% non-fat milk in TBST at room temperature for 1h and incubated with primary antibodies overnight at 4 °C. Secondary antibodies were used at 1:10,000 dilution and incubated at room temperature for 1 h. The membrane was imaged by iBright FL1000. Antibodies used included β-actin (Santa Cruz, Dallas, TX, USA, cat# sc69879), AMPK (Cell Signaling, Danvers, MA, USA, cat# 2532), p-AMPK (Cell Signaling, Danvers, MA, USA, cat# 2535), PDE4B (Novus, Littleton, CO, USA, cat# NB100-2562), anti-rabbit secondary antibody (LICOR, Lincoln, NE, USA, cat# 925-68071), anti-mouse secondary antibody (LICOR, Lincoln, NE, USA, cat# 926-68070). 

### 2.7. Measurement of TNF-α Level

RAW264.7 cells were plated in 6 well plates and were stimulated with 100 ng/mL LPS with or without 20 min pretreatment of 10 μM KVA-D88. The intracellular TNF-α mRNA expression level was determined 6 h after LPS stimulation by real time RT-PCR. The TNF-α level in cell culture medium was determined 24 h after LPS stimulation by ELISA (Cayman Chemical, Ann Arbor, MI, USA, Cat# 500850) according to the instructions. 

### 2.8. Preparation and Characterization of Nanoparticles

Methoxy poly(ethylene glycol)-b-poly(carbonate-co-lactide) [mPEG-b-P(CB-co-LA)] was synthesized as previously described [14]. NPs were prepared by solvent evaporation at room temperature. Briefly, 1 mg KVA-D88 and 10 mg mPEG-b-P(CB-co-LA) copolymer were dissolved in a 2 mL mixture of methylene chloride and acetone (50:50, *v*/*v*), followed by adding 4 mL of 1% polyvinyl alcohol (PVA) aqueous solution. The solution was emulsified by a probe sonicator (Qsonica) for 4 min on ice. The organic solvent was removed by vacuum evaporation. The solution was centrifuged at 5000 rpm for 5 min to remove the unencapsulated drug and the supernatant was collected for characterization of the NPs. The particle size distribution and zeta potential were measured by Malvern Zetasizer (Worcestershire, UK). For in vivo studies, the supernatant was further centrifuged at 20,000 rpm for 30 min to remove PVA and collect the NPs. The NPs were resuspended with PBS and filtered through a 0.2 μm filter. 

### 2.9. Measurement of Drug Loading and Drug Release

KVA-D88 concentration was determined by HPLC using a C_18_ column (150 mm × 4.6 mm, 5 μm, Phenomenex, Torrance, CA, USA). The mobile phase was composed of water: acetonitrile (35: 65, *v*/*v*) and the maximum absorbance was at 296 nm. For drug loading, 100 µL of KVA-D88 formulation was dissolved in 900 μL DMSO followed by 0.2 μm filtration and injected into HPLC. For the drug release study, 1.5 mL of KVA-D88 formulation was placed in the Float-A-Lyzer^®^ (Sigma-Alderich, St. Louis, MO, USA) and dialyzed against 50 mL PBS containing 1% Tween 80 for sink condition. The release study was carried out at 37 °C, 120 rpm on an orbital shaker. At predetermined time points, 50 μL of aliquot was drawn and the release medium was replenished with PBS. The samples were centrifuged at 5000 rpm for 5 min. The supernatant was collected, dissolved in DMSO, and injected into HPLC. All the samples were analyzed in triplicate.

### 2.10. In Vivo Study

All the animal studies were approved by the Institutional Animal Care and Use Committee (IACUC) of the University of Nebraska Medical Center, Omaha, NE, and performed according to the National Institutes of Health (NIH) guidelines. The NIAAA chronic and binge ethanol feeding mice model was used for in vivo study [15]. Briefly, 8- to 10-week-old male C57BL/6 mice were fed with control Lieber-DeCarli diet ad libitum for three days to acclimatize them to tube feeding and liquid diet. Afterward, mice were divided into four groups (*n* = 5) for different treatments: (1) control; (2) EtOH feeding; (3) EtOH feeding with KVA-D88 free drug treatment (free drug group); and (4) EtOH feeding with KVA-D88 NPs treatment (NPs group). The control group was fed with the control Lieber–DeCarli diet ad libitum for 6 weeks. EtOH group, free drug group, and formulation group were fed with ethanol Lieber–DeCarli diet ad libitum containing 5% (*v*/*v*) ethanol for 6 weeks. The treatment groups were intravenously injected with KVA-D88 free drug or NPs (5 mg/kg, twice a week) starting in the third week. At the end of week 6, EtOH feeding group, free drug group, and formulation group were given an oral gavage in the early morning with a single dose of EtOH (5 g/kg). The control group received an oral gavage of 45% isocaloric maltose dextrin in parallel. All four groups were sacrificed 9 h after the oral gavage. Mice blood and major organs were harvested for further analysis.

### 2.11. Measurement of Plasma Levels of Enzymes

Plasma aspartate transaminase (AST) and alanine aminotransferase (ALT) levels were measured by the Comprehensive Diagnostic Profile Kit (cat# 10023238) on a VetScan VS2 (Abaxis North America, Union City, CA, USA) according to the manufacturer’s instructions. Briefly, freshly collected mouse blood was centrifuged at 3000 rpm, 4 °C for 10 min and supernatant plasma was collected and 100 μL plasma was added into the comprehensive diagnostic profile reagent rotor and sent to the Vetscan VS2 chemistry analyzer to measure AST and ALT levels. ALT and AST plasma levels were recorded. 

### 2.12. Measurement of Hepatic Triglyceride Level

Hepatic triglyceride (TG) level was determined by a commercially available kit (Cayman, Ann Arbor, MI, USA, cat# 10010303) according to the manufacturer’s instructions. 

### 2.13. Histological and Immunohistochemistry Analysis

Liver tissues were fixed in 10% paraformaldehyde solution overnight and embedded in paraffin. Hematoxylin and eosin (H&E), Sirius red, and immunohistochemical (IHC) staining was carried out on 5 µm tissues slice following the standard protocols. Antibodies used for IHC included anti-4 hydroxynonenal antibody (Abcam, Cambridge, UK, cat# ab46545) and anti-PDE4B antibody (FabGennix, Frisco, TX, USA, cat# 21-040-CV). For oil red O staining, fresh tissues were embedded in OCT into a plastic cryomold on dry ice and stained with oil red O following the standard protocol.

## 3. Results

### 3.1. KVA-D88 Selectively Inhibits PDE4B Activity

KVA-D88 caused minor toxicity after 48h incubation as determined by MTT assay with murine hepatocyte cell lines AML12 cells and human hepatic stellate cells LX-2 (Figure 1A,B) [16,17]. Cells transfected with PDE4B or PDE4D expression vectors and cAMP response element (CRE) luciferase reporters were used to assess KVA-D88 selectivity [10]. 

Intracellular cAMP binds to CRE and induces luciferase expression. Therefore, the intensity of CRE luciferase is negatively related to PDE4B activity as PDE4B degrades cAMP. In contrast, the intensity of CRE luciferase is positively associated with inhibition efficacy. To investigate the selectivity of KVA-D88 between PDE4B and PDE4D, we also evaluated the effectiveness of KVA-D88 on PDE4D. In Figure 1C, the fold induction implies the intensity of CRE luciferase. The inhibition capability of KVA-D88 showed a concentration-dependent manner on both PDE4B and PDE4D. The CRE luciferase intensity of PDE4B transfected cells is substantially higher than that of PDE4D transfected cells after KVA-D88 treatment ranging from 2.5 to 10 μM, suggesting a lower activity of PDE4B compared to PDE4D. These results demonstrated that KVA-D88 exhibited inhibition and selectivity on PDE4B versus PDE4D. Based on the data, we determined 10 μM as the working concentration, at which KVA-D88 showed good selectivity on PDE4B and imperceptible cytotoxicity as the cell viability remained above 80%. 

It is reported that ALD development is closely related to increased PDE4 expression and decreased cAMP level. We treated murine hepatocytes AML12 cells with 50 mM EtOH and KVA-D88 for 48 h. PDE4B expression was notably upregulated by EtOH, as indicated by both qRT-PCR and Western blot (Figure 1D,E). Although it did not decrease after EtOH incubation, the intracellular cAMP level was dramatically elevated by PDE4B inhibitor KVA-D88 (Figure 1F). Our data indicate that KVA-D88 could selectively inhibit PDE4B activity, prevent cAMP degradation, and upregulate intracellular cAMP level. 

### 3.2. KVA-D88 Represses Inflammatory Responses in Macrophages

Chronic alcohol exposure is known to increase lipopolysaccharide (LPS) leakage from the gastrointestinal tract [18]. LPS is the major component of the outer membrane in gram-negative bacteria and stimulates hepatic Kupffer cells. Activated Kupffer cells and relevant inflammatory response are the dominant factors in ALD progression [19,20]. It is well-studied that PDE4 regulates inflammation via the degradation of cAMP [21]. To determine the anti-inflammatory potential of KVA-D88, macrophage-like RAW 264.7 cells were pretreated with KVA-D88 and stimulated with LPS. The mRNA expression level of PDE4B in macrophages was significantly upregulated by LPS (Figure 2A). The LPS stimulation markedly raised the mRNA expression level of proinflammatory cytokine TNF-α in macrophages, while KVA-D88 treatment decreased TNF-α mRNA level (Figure 2B). ELISA assay revealed that KVA-D88 also suppressed LPS-induced TNF-α secretion from macrophages (Figure 2C). Our results indicate that KVA-D88 exhibited an anti-inflammatory effect by suppressing the secretion of proinflammatory cytokine TNF-α.

### 3.3. PDE4B Inhibition by KVA-D88 Promotes the Expression of Genes and Proteins Related to β-Oxidation and Antioxidants In Vitro

Hepatic lipid accumulation and oxidative stress are the features of ALD [22,23,24,25,26]. To examine whether PDE4B inhibition by KVA-D88 regulates energy metabolism and antioxidants, we treated AML12 cells with 50 mM EtOH for 48 h with or without KVA-D88. AMPK regulates the effects of ethanol on fatty acid metabolism and development of alcoholic fatty liver [27]. The activation and phosphorylation of intracellular energy sensor AMP-activated protein kinase (AMPK) inhibits lipogenesis and promotes lipolysis by modulating the expression of its target genes [28]. EtOH exposure slightly reduced phosphorylation of AMPK, while KVA-D88 markedly enhanced phosphorylation of AMPK (Figure 3A). These results are in agreement with previous reports where ethanol was shown to decrease the activation of AMPK, while constitutive activation of AMPK blocks the effects of ethanol [27]. We further determined the expression level of lipolysis and β-oxidation related genes downstream of AMPK, including peroxisome proliferator-activated receptor alpha (PPAR-α), peroxisome proliferator-activated receptor gamma coactivator 1-alpha (PGC-1α), carnitine palmitoyltransferase 1A (CPT-1A), and peroxisomal acyl-coenzyme A oxidase 1 (ACOX1). The transcription factor PPAR-α and transcription coactivator PGC-1α downstream of AMPK are major regulators of energy homeostasis. Induction of PPAR-α and PGC-1α by p-AMPK upregulates the expression of CPT-1A and ACOX1, which catalyzes fatty acid β-oxidation and inhibits lipolysis [29,30]. Although EtOH exposure for 48h did not significantly alter the expression level of these genes, KVA-D88 significantly elevated their expression (Figure 3B). We also found that KVA-D88 upregulated the expression of two antioxidative enzymes, superoxide dismutase 1 and 2 (SOD1 and SOD2) (Figure 3C). These data indicate that KVA-D88 could promote β-oxidation and protect cells from oxidative stress by modulating relevant gene expression.

### 3.4. KVA-D88 Inhibits the Expression of Profibrotic Genes and Relative Protein

Continuous hepatic injury due to excessive alcohol consumption results in liver fibrosis via activation of HSCs [31]. To determine the anti-fibrotic effect of KVA-D88, we stimulated LX-2 cells with 2.5 ng/mL TGF-β1 with or without KVA-D88. TGF-β1 stimulated LX-2 cells showed increased collagen I (Col I) and α-SMA expression. KVA-D88 suppressed TGF-β1 induced collagen and α-SMA expression at mRNA levels as determined by real time RT-PCR (Figure 4A,B) and protein levels as determined by Western blot analysis (Figure 4C,D). Western blots were also quantified and piloted in Figure 4E,F. 

### 3.5. Preparation and Characterization of KVA-D88 Loaded Nanoparticles

KVA-D88 is a small hydrophobic molecule with the ability to cross the blood–brain barrier (BBB) [10]. Inhibition of PDE4D in the central nervous system is closely related to gastrointestinal side effects, as suggested by other PDE4 inhibitors [7,32,33]. Therefore, to avoid the risk of gastrointestinal side effects, improve the solubility and maximize the biodistribution to the liver, we encapsulated KVA-D88 into the NPs prepared using mPEG-b-P(CB-co-LA) copolymer (Figure 5A). The mean particle size of NPs was 135 ± 10 nm with the polydispersity index (PDI) of 0.146. Further, NPs were stable for 3 days in PBS as determined by DLS (Figure 5B). The zeta potential was found close to neutral, as the polymer is devoid of any charged moiety, and the drug encapsulation is accomplished by the hydrophobic interactions only (Figure 5C). The copolymer could efficiently load the hydrophobic compound KVA-D88 and the drug loading was 4.2% *w*/*w* as determined by HPLC. KVA-D88-loaded NPs provided a sustained drug release profile (Figure 5D). The drug-loaded NPs displayed a burst release of 15% within the first 3 h and a sustained release of 60% after 48 h, followed by a slower and continuous release till 120 h. These data indicate that KVA-D88-loaded mPEG-b-P(CB-co-LA) NPs were stable and performed a sustained release profile.

### 3.6. KVA-D88 Loaded Nanoparticles Alleviate Alcohol-Induced Liver Injury and Inflammation In Vivo

Having confirmed the potential of KVA-D88 to inhibit PDE4B activity and suppress inflammation in vitro, we tested the efficacy of KVA-D88 loaded NPs in mice using the NIAAA model that is suitable for studying ALD pathogenesis and alcohol-related damage [6,15,34,35]. Alcohol consumption for 6 weeks significantly elevated the hepatic expression level of PDE4B in mice, as indicated by IHC (Figure 6A). The plasma levels of alanine transaminase (ALT) and aspartate aminotransferase (AST) were markedly increased by alcohol feeding, while KVA-D88-loaded NPs significantly reduced alcohol-induced upregulation of ALT and AST (Figure 6C,D) [36,37]. H&E staining showed that chronic alcohol exposure caused severe liver injury and steatosis, which were ameliorated by KVA-D88 loaded NPs as indicated by improved hepatic architecture and fewer lipid droplets accumulation (Figure 6B). Hepatic inflammation induced by LPS leakage accelerates the progression of ALD. Alcohol exposure resulted in a substantial increase in hepatic proinflammatory cytokine levels. Administration of KVA-D88 loaded NPs showed pronounced anti-inflammatory efficacy in vivo as indicated by a significant decrease in TNF-α and IL-1β mRNA expression (Figure 6E,F). These data indicate that KVA-D88 could improve alcohol-induced liver injury and suppress inflammation in response to continuous alcoholic intake, which is consistent with in vitro data. 

### 3.7. KVA-D88 Loaded Nanoparticles Alleviate Alcohol-Induced Steatosis and Promote the Expression of Genes Involved in β-Oxidation In Vivo

Steatosis is one of the most common and characterized stages during ALD [25]. Oil red staining estimated that alcohol feeding resulted in acute steatosis with extensive deposition of oil droplets in the liver. KVA-D88 loaded NPs dramatically reduced hepatic lipid accumulation to the level of the control group (Figure 7A). Triglyceride (TG), also known as triacylglycerol (TAG), is the principal constituent of fat accumulated in the liver [6,35,38]. The level of hepatic TAG further confirmed that alcohol feeding induced steatosis while KVA-D88 loaded NPs significantly ameliorated alcohol-induced fat aggregation (Figure 7B). It is reported that chronic alcohol intake leads to steatosis by promoting hepatic lipogenesis and suppressing fatty acid β-oxidation [25,39]. To investigate whether KVA-D88 could promote β-oxidation in vivo, we measured the expression level of PPAR-α, PGC-1α, CPT-1A, and ACOX1 [40,41,42]. Incoherent with other studies, alcohol notably downregulated the expression of these genes. However, the downregulation of PPAR-α, PGC-1α, CPT-1A, and ACOX mediated by alcohol was greatly reversed by KVA-D88 loaded NPs (Figure 7C–F). These data suggest that KVA-D88 NPs could effectively ameliorate alcohol-related steatosis via promoting the expression of key genes regulating hepatic β-oxidation. 

### 3.8. KVA-D88 Loaded Nanoparticles Alleviate Alcohol-Induced Oxidative Stress and Fibrosis

The metabolism of excessive alcohol is inevitably associated with oxidative stress [24,43]. The oxidants generated by alcohol metabolism attack biomolecules and damage the functional structures [23]. Lipid peroxidation is an acknowledged consequence of oxidative stress during ALD which is characterized by oxidative degradation of lipids in the cellular membrane and results in impaired functions [44]. Malondialdehyde (MDA) and 4-hydroxynonenal (4-HNE) are the two major products of lipid peroxidation [6,45,46]. Chronic alcohol diet significantly decreased the expression of antioxidant enzymes, SOD1 and SOD2. While KVA-D88 loaded NPs prevented the downregulation of SOD1 and SOD2 induced by ethanol (Figure 8A,B). To evaluate the damages caused by oxidative stress, we assessed the hepatic level of 4-HNE by IHC. Alcohol feeding markedly elevated the hepatic content of 4-HNE, which was decreased by KVA-D88 loaded NPs (Figure 8C). Oxidative stress, inflammation, and chronic liver injury during ALD lead to the occurrence of fibrosis [31]. Sirius red staining of mouse liver showed that chronic alcohol consumption resulted in fibrosis, which was ameliorated by KVA-D88 loaded NPs (Figure 8D). These data prove that KVA-D88 loaded NPs could improve alcohol-mediated oxidative stress and fibrosis. 

## 4. Discussion

ALD is a significant health problem caused by excessive alcohol consumption. ALD is generally associated with fatty liver, hepatitis or fibrosis progression, leading to cirrhosis [47]. Unfortunately, there is no approved drug for ALD, and the current therapies rely on abstinence, immunosuppressants like corticosteroids or liver transplantation in case of advanced cirrhosis [2]. Excess use of corticosteroids can cause chronic inflammatory diseases (asthma, chronic obstructive pulmonary disease, inflammatory bowel disease), while access to liver transplantation is limited as the demand for organs exceeds availability [48,49]. Therefore, therapeutic interventions are immediately needed for ALD. 

Studies suggest that inflammation plays a significant role in ALD initiation and progression [50]. Kupffer cells (KCs) and monocytes induced macrophages in the liver are mainly involved in hepatic inflammatory reactions [19,20]. KCs, upon contact with injured hepatocytes or gut bacterial endotoxins such as LPS, are activated and secrete inflammatory cytokines, including TNF-α and start the inflammation [51]. Pharmacological inhibition of KCs by small molecules, such as gadolinium chloride or decreasing intestinal bacterial load with antibiotics, has been shown to prevent liver injury and ALD progression [52,53]. 

cAMP is an intracellular second messenger that plays an important role in ALD progression [4,6]. The binding of cAMP to PKA regulatory subunits activates its catalytic unit, phosphorylating downstream protein targets. PKA substrates regulate lipid and glucose metabolism, smooth muscle contraction, and more biological activities [4]. Activated PKA leads to increased glucose production by increasing the transcription of gluconeogenic enzymes, glucose 6-phosphatase (G6Pase), phosphoenolpyruvate carboxykinase (PEPCK) and pyruvate carboxylase (PC). Further, activated PKA subunits inhibit lipogenesis through phosphorylation and inhibition of the key enzymes such as acetyl-CoA carboxylase (ACC) and pyruvate dehydrogenase (PD) [54]. Activated PKA also affects lipid metabolism through activation of cAMP-responsive transcription factors such as the cAMP response element-binding protein (CREB) [55]. On the other hand, ethanol causes defects in the nuclear translocation and phosphorylation of CREB, leading to increased lipid accumulation in the liver and suppression of carnitine palmitoyltransferase 1A (CPT1A), the rate limiting enzyme for mitochondrial fatty acid oxidation. ALD patients often display metabolic disorder accompanied by lower cAMP levels in the liver [56]. Furthermore, ethanol, by affecting the tight junction protein, causes a leaky gut and endotoxemia [57]. LPS-induced TNF-α expression in KCs plays a key role in alcoholic hepatitis and ALD. Studies have demonstrated that inhibiting TNF-α activity with excessive cAMP in vitro or by anti-TNF-α antibody administration in vivo shows liver protective effects [58,59]. Therefore, cAMP signaling modulation has been identified as a new strategy for treating ALD, cirrhosis, and HCC. 

Chronic alcohol abuse upregulates the expression of cAMP-degrading enzyme PDE4 in patients and suppressing PDE4 has been shown to improve ALD [6]. In our study, we found that alcohol exposure increased PDE4B mRNA and protein levels (Figure 1D,E). Adenylyl cyclase (AC) activity is increased in the presence of ethanol by enhancing the agonist (prostaglandin E1) stimulation [60]. Furthermore, the cAMP system is known to activate PDE4 gene expression for self-regulation in different cell types [61]. Consequently, we observed a significant increase in PDE4B mRNA expression upon KVA-D88 mediated upregulation of intracellular cAMP (Figure 2A). PDE4 inhibitors are well-studied and marketed as anti-inflammatory therapeutics for skin and lung diseases [62]. PDE4 inhibitors have been shown to regulate alcohol intake without altering its metabolism, showing that PDEs also play a role in reward and motivational behavior [63]. Two of the PDE4 inhibitory drugs, Roflumilast and Apremilast, are approved for treating chronic obstructive pulmonary disease and psoriasis, respectively. However, inhibition of pan PDE4 results in severe side effects and use of these drugs is limited to treating moderate to severe conditions [7,32]. In a clinical trial of Roflumilast, 67% patients and 19% patients treated with Roflumilast reported adverse effects and serious adverse effects, respectively. Further, 14% of Roflumilast group patients discontinued the clinical study and their discontinuations were associated with adverse effects, including diarrhea, nausea, and headache. Vomiting has also been reported. Patients in the Roflumilast group with diarrhea, nausea, vomiting, or headache showed more weight loss compared to the patients without adverse events [64]. It was revealed that these side effects are induced mainly by inhibition of PDE4D isotype, specifically in the central nervous system [8]. To address this issue, we utilized a novel small molecule PDE4B specific inhibitor KVA-D88 [9]. In the cell-based assay, our group has shown that KVA-D88 is 10 time more potent in increasing the cAMP levels with EC 50 of 500 nM compared to Apremilast [10]. Our results also show that KVA-D88 has low cytotoxicity as cell viability remained above 80% after 48 h incubation of AML12 and LX-2 cells at the concentrations up to 20 μM (Figure 1A,B). We showed that KVA-D88 was able to increase the intracellular cAMP concentrations and showed stronger inhibition of PDE4B over PDE4D (Figure 1C,D). 

To better mimic the pathology of ALD, we have applied the NIAAA mouse model to induce PDE4B expression and a series of symptoms of ALD. This model can also elevate blood alcohol levels and AST and ALT levels compared to chronic or binge feeding alone [15] (Figure 6C,D). Plasma ALT and AST levels and H&E staining indicated severe liver injury after alcohol feeding. PDE4B inhibition by KVA-D88 notably ameliorated the damages, and the NPs showed better efficacy compared to the free drug KVA-D88 (Figure 6E,F).

Steatosis is the most common consequence of excessive alcohol consumption and around 90% of heavy drinkers are diagnosed with steatosis [25,65]. In the hepatocytes, free fatty acid (FFA) is degraded via β-oxidation for energy supply or converted to triglycerides (TG) via lipogenesis and excreted by very-low-density lipoprotein (VLDL). While excessive TG may be stored in hepatocytes as lipid droplets and results in steatosis [66,67]. Our study showed that PDE4B inhibition by KVA-D88 markedly improved alcohol-induced steatosis in mice as indicated by Oil red O staining and hepatic TG content (Figure 7A,B). The level of p-AMPK/AMPK, PPAR-α, PGC-1α, CPT-1A, and ACOX1 were also reversed by KVA-D88 after alcohol exposure in vivo and in vitro (Figure 3A,B and Figure 7C–F). Since AMPK activation is known to regulate both β-oxidation and lipogenesis [68], this could have prevented an ethanol-mediated decrease in major antioxidant enzymes, cytosolic and mitochondrial SOD1/2, which would alleviate ROS production by ethanol metabolism and formation of acrolein adducts. 

Inflammatory response protects the body from possible harmful stimuli while the continuous and overwhelming inflammation caused by alcohol abuse is the first hit during ALD and accelerates the progression of the disease [18]. Therefore, we assessed the anti-inflammatory efficacy of KVA-D88. Our in vitro data indicated that KVA-D88 could inhibit the LPS induced production of inflammatory cytokine TNF-α in macrophages (Figure 2). The in vivo study further confirmed this conclusion as KVA-D88 treatment alleviated alcohol-induced inflammation in mice (Figure 6E,F). Activation of HSCs indicated by α-SMA and excess secretion of extracellular matrix (ECM) proteins including collagen is a hallmark of liver fibrosis [69]. cAMP/PKA pathway has a direct role in this phenotype of HSCs through CREB activation. In quiescent HSCs, CREB is phosphorylated at serine 133, which decreases upon HSC activation [70]. It is known that activation of PKA or CAMK-II restores phospho-CREB levels and inhibits proliferation of activated HSCs [4]. Our results proved that PDE4B inhibition by KVA-D88 exhibited an anti-fibrotic efficacy as KVA-D88 treatment decreased the mRNA and protein expression of Col 1A1 and α-SMA in vitro and alleviated fibrosis in vivo (Figure 4 and Figure 8D).

The gastric side effects of PDE4 inhibition could also be reduced by regulating drug distribution in the body. Several approaches, such as inhalation or topical routes of administration, antibody drug conjugation and nanoformulations, have been applied by others to limit the exposure of PDE4 inhibitors to the tissue or cell of interest [71,72]. However, while these approaches could be effective for targeting certain organ systems, they are extendable to internal organs or require complex formulation. Since KVA-D88 is water insoluble, we used mPEG-b-P(CB-co-LA) to prepare NPs to encapsulate this drug for its sustained release and enhanced delivery to the liver after systemic administration. The mPEG corona provides stealth property and allows longer circulation time in vivo [73]. KVA-D88 loading into the NPs was 4.2 ± 0.5%, with an average particle size of 135 ± 10 nm and PDI less than 0.2 (Figure 5B). NPs maintained a continuous release profile of KVA-D88 for 120 h in vitro (Figure 5D). We have previously shown that these NPs accumulate preferentially in the liver and, due to the biodegradable nature of the polymer, they are completely harmless [37,74,75]. The NP approach could also avoid any possible CNS side effects associated with this drug.

Oxidative stress induced by alcohol metabolism plays a significant role in cellular damage [22]. Three enzymes catalyze the oxidation of EtOH: alcohol dehydrogenase (ADH), cytochrome P450 2E1 (CYP2E1), and peroxisomes. CYP2E1 is the major inducible enzyme, and its activation generates large amounts of ROS [47]. The highly oxidative species forms adducts with proteins and lipids, disrupting the biological structures and functions [76]. We found that KVA-D88 had antioxidative efficacy as it elevated the level of antioxidants SOD1 and SOD2 (Figure 3C). Our in vivo data supported this finding that KVA-D88-treated mice showed lower hepatic content of 4-HNE, a lipid peroxidation product (Figure 8C). HSCs activated by TGF-β1 are the major contributor to collagen deposition during liver fibrosis [77]. Inhibition of HSCs activation can significantly improve liver fibrosis [78]. Our study found that KVA-D88 reduced TGF-β-stimulated collagen and α-SMA production in vitro (Figure 4E,F).

## 5. Conclusions

We have utilized a novel PDE4B inhibitor KVA-D88 as a therapeutic agent for ALD. KVA-D88 showed higher inhibitory action on the PDE4B isotype of enzymes, increased intracellular cAMP levels, suppressed inflammatory response and increased the expression of key genes involved in β-oxidation in vitro. Further, liver targeted delivery of KVA-D88-loaded NPs significantly improved alcohol-induced hepatic injury, inflammation, steatosis, oxidative stress, and fibrosis in mice. In conclusion, our results support that KVA-D88-loaded NPs are promising for ALD treatment and offer a new strategy for ALD therapy. 

## Figures and Tables

**Figure 1 pharmaceutics-14-01894-f001:**
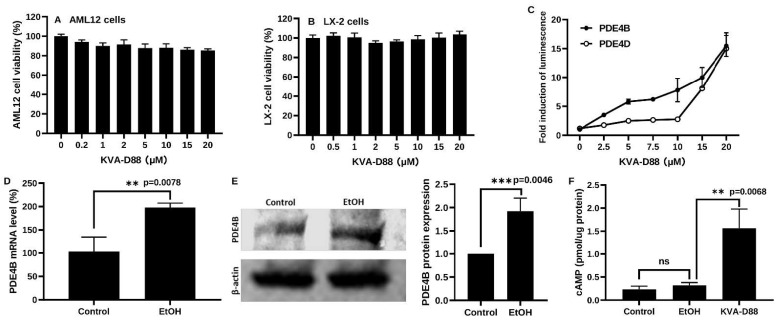
KVA-D88 selectively inhibits PDE4B activity. (**A**,**B**) Cell viability of AML 12 and LX-2 cells after 48 h treatment with KVA-D88 as determined by MTT assay. (**C**) Dose-dependent activities of PDE4B and PDE4D after treatment with KVA-D88 as determined by CRE luciferase assay. The fold induction of luminescence indicated the inhibitory efficacy. (**D**,**E**) PDE4B mRNA and protein expression in AML 12 cell line after 50 mM EtOH treatment for 48 h. (**F**) Intracellular cAMP level after 50 mM EtOH treatment for 48 h with or without 10 μM KVA-D88. Results are presented as the mean ± S.D. (*n* = 3). ** *p* < 0.01, *** *p* < 0.005, and ns *p* > 0.05.

**Figure 2 pharmaceutics-14-01894-f002:**
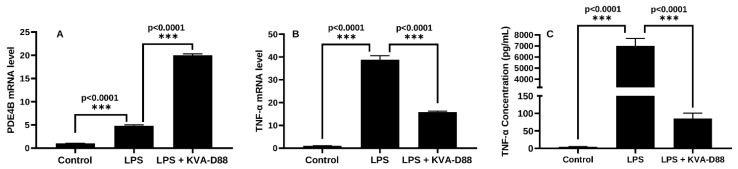
KVA-D88 represses inflammatory responses in macrophages. RAW264.7 cells were stimulated with 100 ng/mL LPS with or without 20 min pretreatment of 10 μM KVA-D88. mRNA expression of PDE4B (**A**) and TNF-α (**B**) was determined by real time RT-PCR 6 h after LPS stimulation. (**C**) TNF-α level in the culture medium was measured by ELISA assay 24 h after LPS stimulation. Results are presented as the mean ± S.D. (*n* = 3). *** *p* < 0.005.

**Figure 3 pharmaceutics-14-01894-f003:**
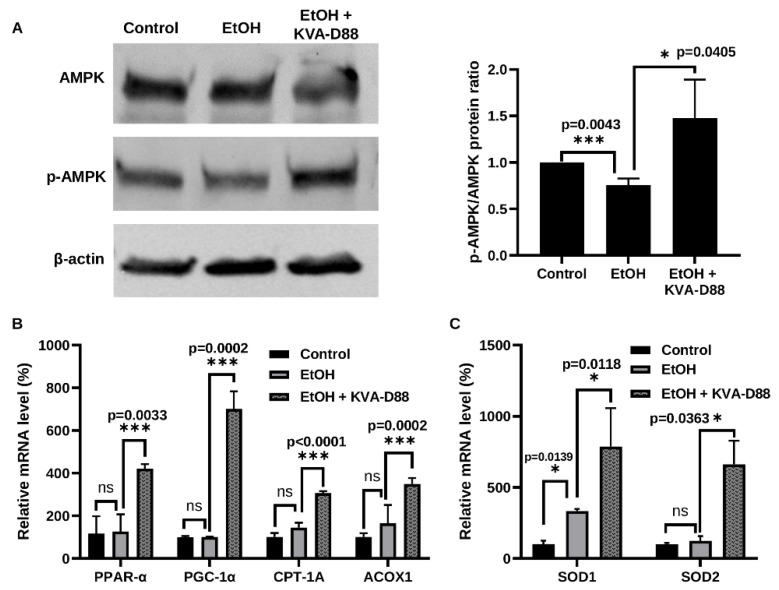
KVA-D88 promotes the expression of genes and proteins related to β-oxidation and antioxidants. AML 12 cells were treated with 50 mM EtOH for 48 h with or without 10 μM KVA-D88. (**A**) Western blot of AMPK and p-AMPK. (**B**) mRNA expression level of PPAR-α, PGC-1α, CPT-1A, and ACOX. (**C**) mRNA expression level of SOD1 and SOD2. Results are presented as the mean ± S.D. (*n* = 3). * *p* < 0.05, *** *p* < 0.005, and ns *p* > 0.05.

**Figure 4 pharmaceutics-14-01894-f004:**
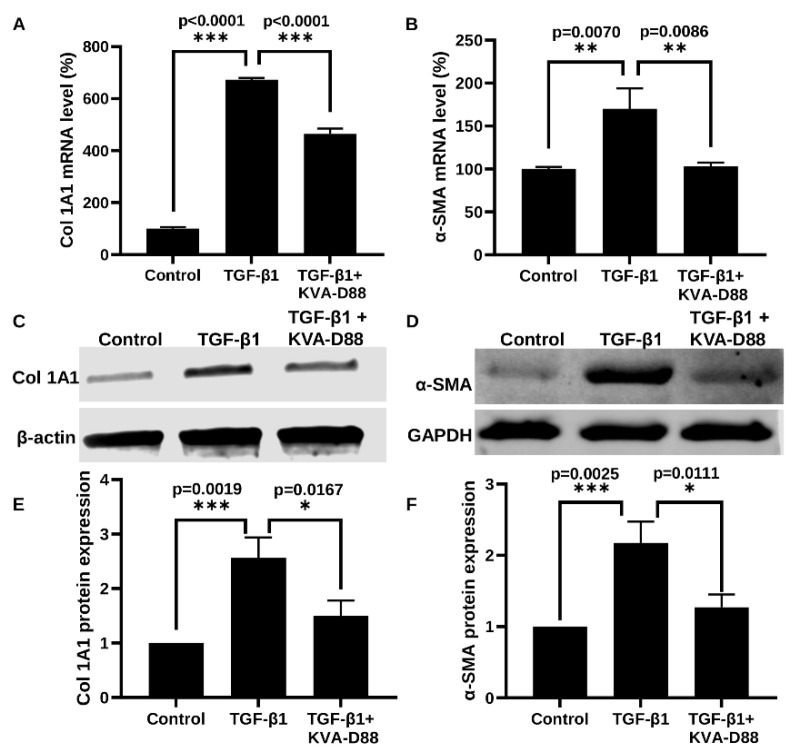
KVA-D88 inhibits profibrotic genes and relative protein expression. LX-2 cells were stimulated with 2.5 ng/mL TGF-β1 for 24 h with or without 20 min preincubation of 10 µM KVA-D88. (**A**,**B**) Col 1A1 and α-SMA mRNA expression as determined by real time RT-PCR. (**C**,**D**) Collagen 1A1 and α-SMA protein expression as determined by Western blot analysis and their quantification of blot bands (**E**,**F**). Results are presented as the mean ± S.D. (*n* = 3). * *p* < 0.05, ** *p* < 0.01 and *** *p* < 0.005.

**Figure 5 pharmaceutics-14-01894-f005:**
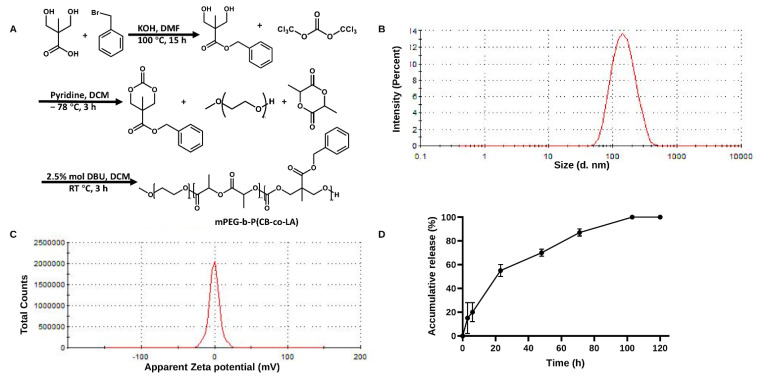
Preparation and characterization of KVA-D88 loaded nanoparticles. (**A**) Synthesis of mPEG-b-P(CB-co-LA). (**B**) Particle size distribution of KVA-D88 loaded nanoparticles. (**C**) Zeta potential of KVA-D88 loaded nanoparticles. (**D**) Release profile of KVA-D88 loaded nanoparticles in PBS. Results are presented as the mean ± S.D. (*n* = 3).

**Figure 6 pharmaceutics-14-01894-f006:**
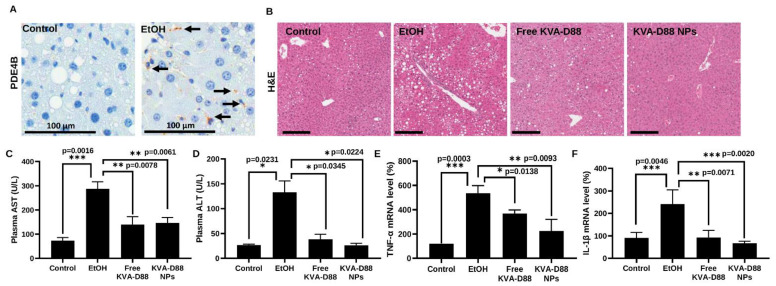
KVA-D88 loaded nanoparticles alleviate alcohol-induced liver injury and inflammation after systemic administration into mice. (**A**) Representative microscopic pictures of IHC staining for PDE4B. PDE4B expression was upregulated in mice liver by alcohol diet. Scale bars (120×), 100 μm. (**B**) Representative microscopic pictures of H&E staining of mice liver tissue. Scale bars (20×), 200 μm. (**C**,**D**) Mice plasma AST and ALT levels. (**E**,**F**) Mice hepatic mRNA expression level of TNF-α and IL-1β. Results are presented as the mean ± SD (*n* = 3). * *p* < 0.05, ** *p* < 0.01 and *** *p* < 0.005.

**Figure 7 pharmaceutics-14-01894-f007:**
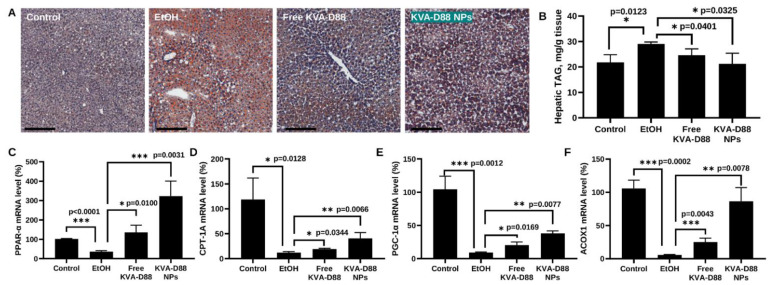
KVA-D88 loaded nanoparticles alleviate alcohol-induced steatosis and promote β-oxidation related gene expression in the liver after systemic administration into mice. (**A**) Representative microscopic pictures of oil red O staining of mice liver tissue. (**B**) Mice hepatic triglyceride (TG) level. Mice hepatic mRNA expression levels of PPAR-α (**C**), CPT-1A (**D**), PGC-1α (**E**), and ACOX1 (**F**). Scale bars (20×), 200 μm. Results are presented as the mean ± SD (*n* = 3). * *p* < 0.05, ** *p* < 0.01 and *** *p* < 0.005.

**Figure 8 pharmaceutics-14-01894-f008:**
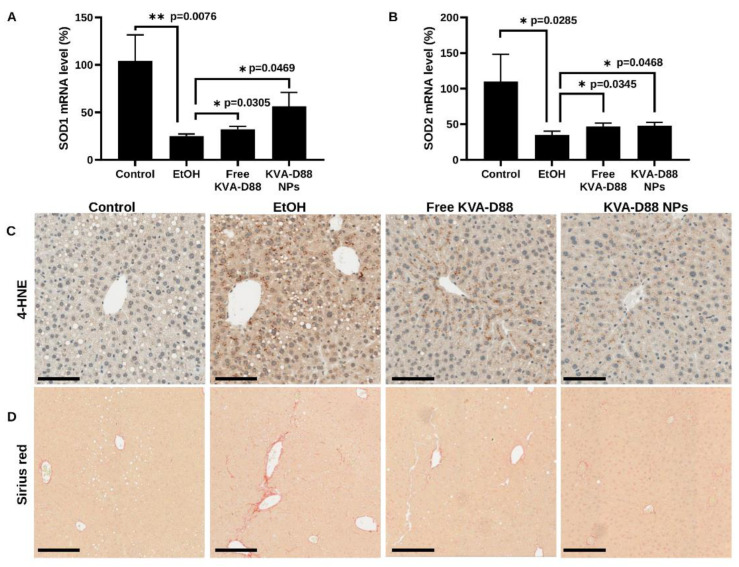
KVA-D88 loaded nanoparticles alleviate alcohol-induced oxidative stress and fibrosis after systemic administration into mice. (**A**,**B**) Mice hepatic mRNA expression levels of SOD1 and SOD2. Results are presented as the mean ± SD (*n* = 3). * *p* < 0.05 and ** *p* < 0.01. (**C**) Representative microscopic pictures of IHC staining for 4-HNE. (**D**) Representative microscopic pictures of Sirius red staining of mice liver tissue. Scale bars (20×), 200 μm.

## Data Availability

We cited one data from https://wonder.cdc.gov/.

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
