# Peer review of "Nanoparticle Delivery of Novel PDE4B Inhibitor for the Treatment of Alcoholic Liver Disease"

_pharmaceutics, 2022, doi:10.3390/pharmaceutics14091894_

Round 1

Reviewer 1 Report

The present manuscript entitled “Nanoparticle delivery of novel PDE4B inhibitor for the treatment of alcoholic liver disease” is an extensive study covering many aspects to prove the hypothesis. The study design is appropriate and covers all the studies needed to justify the title. Few below comments can be considered/justified before the final acceptance of the manuscript. 

1) Line 166, please include the proper citation rather than doi (if available).

2) Error bars are missing in figure 1 E.

3) Authors are advised to double-check the statistical analysis significances in figure 2A between control and LPS, LPS+KVA-D88, and figures 3, and 4. Although the authors mentioned statistical significance is p˂0.0001 in all the cases, however, observation tells a little different story.

4) Authors have not shown any entrapment efficiency data, even though they have performed the study.

5) I am just curious why the KVA-D88 nanoparticles were tested only for in-vivo studies not for in-vitro studies.

Author Response

Reviewer #1

The present manuscript entitled “Nanoparticle delivery of novel PDE4B inhibitor for the treatment of alcoholic liver disease” is an extensive study covering many aspects to prove the hypothesis. The study design is appropriate and covers all the studies needed to justify the title. Few below comments can be considered/justified before the final acceptance of the manuscript. 

  1. Line 166, please include the proper citation rather than doi (if available).

We have made corrections according in the text.

  1. Error bars are missing in Figure 1E.

When plotting the figure, we added the error bar in the treatment group but not the control group. This is due to the method we used to quantify the result of western blot. When quantifying the band density with Image J, we set the band density in the control groups as 1 and then calculated the fold change of treatment groups. Therefore, the fold change of bands in the control group is 1, without a standard deviation or error bar.

The same method can be seen in other references:

  • Hart, Martin, et al. "miR-34a as hub of T cell regulation networks." Journal for immunotherapy of cancer 7.1 (2019): 1-11.;
  • Yi, Juan, et al. "Desflurane preconditioning induces oscillation of NF-κB in human umbilical vein endothelial cells." PloS one 8.6 (2013): e66576;
  • You, Yan, et al. "SNX10 mediates alcohol-induced liver injury and steatosis by regulating the activation of chaperone-mediated autophagy." Journal of hepatology 69.1 (2018): 129-141.
  1. Authors are advised to double-check the statistical analysis significances in Figure 2A between control and LPS, LPS + KVA-D88, and Figures 3, and 4. Although the authors mentioned statistical significance is p˂0.0001 in all the cases, however, observation tells a little different story.

We checked the PCR data in Figure 2A and confirmed that the p-value is less than 0.0001 as indicated in the figure. Each group was performed in triplicate, and comparisons between two different groups were analyzed by student’s t-test. The result was copied as follows:

mRNA expression fold change

Control

LPS

LPS + KVA-D88

1.030

4.868

19.744

0.995

5.005

20.019

0.975

4.574

20.299

The p-value between groups is as follows:

Groups

p-value

Control & LPS

7.607E-06

LPS & KVA-D88

1.965E-07

Control & KVA-D88

3.083E-08

  1. Authors have not shown any entrapment efficiency data, even though they have performed the study.

According to the comments, we added the entrapment data to the result at line 311.

  1. I am just curious why the KVA-D88 nanoparticles were tested only for in-vivo studies not for in-vitro studies.

We applied the NPs formulation in vivo for several reasons. (1 to improve the solubility and bioavailability in vivo. For the in vitro study, KVA-D88 was dissolved in DMSO and added to the cell culture medium. The DMSO solution could deliver KVA-D88 free drug to the cells. Therefore, we did not use the NPs formulation when treating cells in vitro. (2 to decrease the biodistribution of KVA-D88 in brain and increase in liver. Since in vitro we only treat one cell line, it is not practicable to test biodistribution in vitro. 3) to provide a continuous release profile. For this purpose, we tested the release profile of NPs in vitro and found that NPs showed a sustained release for up to 120 hours. 

Reviewer 2 Report

The research work "Nanoparticle delivery of novel PDE4B inhibitor for the treatment of alcoholic liver disease" is novel and connected to social cause. 

Comments are as follows: 

1. Cite the procedure with appropriate research work if followed available procedure, like, Cytotoxicity assay, Measurement of KVA-D88 inhibitory activity in vitro etc. 

2. Is KVA-D88, approved or clinically proven safe and effective?

3. Line 166, replace doi with citation. 

4. Is Methoxy poly(ethylene glycol)-b-poly(carbonate-co-lactide) [mPEG-b-P(CB-co-LA)], hydrophobic or hydrophilic polymer?

5. Line 166-167: solvent evaporation, can author specify the conditions. Is drug stable at temperature used?

6. in vivo study, cite at respective places, if reference method used

7. 210 and 214, manufacturer’s instructions, can author elaborate?

8. Line 696: Ref: 66, What is PT? Please correct, after review. 

Author Response

Reviewer#2

The research work "Nanoparticle delivery of novel PDE4B inhibitor for the treatment of alcoholic liver disease" is novel and connected to social cause.

Comments are as follows: 

  1. Cite the procedure with appropriate research work if followed available procedure, like, Cytotoxicity assay, Measurement of KVA-D88 inhibitory activity in vitro etc.

We have added references at line 233 and line 235, which we have followed for cytotoxicity assay and PDE4 activity assay.

  1. Is KVA-D88, approved or clinically proven safe and effective?

KVA-D88 is not yet approved or proven safe in the clinic. In this study, we have shown it has low toxicity in vitro and in animals. The other studies also utilized KVA-D88 in mice and confirmed its safety. Further studies, like PK/PD studies, need to be carried out before KVA-D88 can be tested in the clinical trial.

Reference: KVA-D-88, a novel preferable phosphodiesterase 4B inhibitor, decreases Cocaine-mediated reward properties in vivo    

  1. Line 166, replace doi with citation.

Corrections were made in the text.

  1. Is methoxy poly(ethylene glycol)-b-poly(carbonate-co-lactide) [mPEG-b-P(CB-co-LA)], hydrophobic or hydrophilic polymer?

mPEG-b-P(CB-co-LA) is an amphiphilic copolymer. While mPEG is hydrophilic componenet, the rest component of this polymer is hydrophobic. mPEG part forms a hydrophilic corona on the NP  surface, which can improve solubility and prevent plasma protein adsorption. The hydrophobic component of this copolymer forms the core of the NPs, which can entrap hydrophobic compounds inside NPs via the hydrophobic interactions.

  1. Line 166-167: solvent evaporation, can author specify the conditions. Is drug stable at temperature used?

We have added the condition in the text line 166 according to the comments. NPs were prepared at room temperature, except for sonication, which was performed on ice. KVA-D88 is stable at room temperature for at least 2 weeks. 

  1. in vivo study, cite at respective places, if reference method used.
  • We have added several references that we followed.
  • At line 329, references using the NIAAA mouse model were added.
  • At line 333, references using ALT and AST levels were added.
  • At line 350, references measuring hepatic TAG levels were added.
  • At line 355, references measuring the mRNA expression of relative genes were added.
  • At line 368, references measuring 4-HNE were added.

  1. Lines 210 and 214, manufacturer’s instructions, can author elaborate?

We have added more details in the text from line 207. Serum ALT and AST levels were d using the Comprehensive Diagnostic Profile Kit on the VetScan VS2 Chemistry Analyzer. After centrifuging the whole blood and collecting supernatant, we added 100 μL of plasma into the Comprehensive Diagnostic Profile rotor. Then, we put the rotor into the VetScan VS2 Chemistry Analyzer. After choosing the animal species on the analyzer, the analyzer automatically analyzes the chemistry profiles of plasma samples. The analyzer generated  blood chemistry profiles, including the level of ALT, AST.

  1. Line 696: Ref: 66, What is PT? Please correct, after review.

Correction made according to comments.

Reviewer 3 Report

1. Well designed study and results to support their purposes. However, Please add some descriptions about side effects of PDE4D and thus why using PDE4B in this present study.

2. PDE4B is a novel chemical? Suggesting be careful using :novel!

3. Suggesting the authors have to compare this PDE4B with other similar chemicals or clinical used chemicals /drugs to demonstrate PDE4B is potential!

4. Why the authors do not measure r-GT? 

Author Response

Reviewer#3

  1. Well designed study and results to support their purposes. However, Please add some descriptions about side effects of PDE4D and thus why using PDE4B in this present study.

We have added detailed description about the side effects of a PDE4 inhibitor Roflumilast in the Discussion section, which can represent the typical adverse events of PDE4D inhibition. 

  1. PDE4B is a novel chemical? Suggesting be careful using:novel!

Yes, PDE4B inhibitor KVA-D88 is a novel drug which was synthesized by Dr. Hopkins. They synthesized a series of 1H-pyrrolo[2,3-b]pyridine-2-carboxamide series of PDE4B inhibitors. Among them, KVA-D88 showed the best efficacy. The details of their work can be found in the following paper: Synthesis and SAR Studies of 1H‑Pyrrolo[2,3‑b]pyridine-2carboxamides as Phosphodiesterase 4B (PDE4B) Inhibitors

  1. Suggesting the authors have to compare this PDE4B with other similar chemicals or clinical used chemicals /drugs to demonstrate PDE4B is potential!

This is a good suggestion. (1) Roflumilast and Apremilast are approved PDE4 inhibitors for treating chronic obstructive pulmonary disease and psoriasis, respectively. Their efficacies are based on the anti-inflammatory effect due to PDE4 inhibition. PDE4 is a well-studied target to inhibit inflammation. Many studies have proved that PDE4B is a potential anti-inflammation target in different disease models by inhibiting PDE4B gene expression.

Reference:

Nonredundant Function of Phosphodiesterases 4D and 4B in Neutrophil Recruitment to the Site of Inflammation. 2)Blockade of PDE4B limits lung vascular permeability and lung inflammation in LPS-induced acute lung injury. 3)Inhibition of PDE4/PDE4B improves renal function and ameliorates inflammation in cisplatin-induced acute kidney injury.

A study using PDE4 inhibitor Rolipram demonstrated that PDE4 is a potential target for alcoholic liver disease treatment. Therefore, it inspired us to determine the efficacy of PDE4B inhibitor KVA-D88.

Reference: Phosphodiesterase 4 Inhibition as a Therapeutic Target for Alcoholic Liver Disease: From Bedside to Bench

  1. Why the authors do not measure r-GT? 

This is a good point as the GT level is a biomarker for alcohol abuse. The level of GT is significantly correlated with alcohol intake. However, we induced alcoholic liver disease in mice by giving a diet containing excessive alcohol. Therefore, the GT level in all alcohol-treated groups is supposed to increase. ALT and AST levels are closely related to hepatic damage, which should increase after alcohol abuse and decrease after treatment since the treatment protects the liver from alcohol-induced damage. Compared with GT, ALT and AST levels are better markers to evaluate the efficacy of the drug. Therefore, we measured AST and ALT levels in this study. 
